# Cuminal Inhibits *Trichothecium roseum* Growth by Triggering Cell Starvation: Transcriptome and Proteome Analysis

**DOI:** 10.3390/microorganisms8020256

**Published:** 2020-02-14

**Authors:** Zhong Zhang, Wenting Zhang, Yang Bi, Ye Han, Yuanyuan Zong, Dov Prusky

**Affiliations:** 1College of Food Science and Engineering, Gansu Agricultural University, Lanzhou 730070, China; 2Department of Postharvest Science of Fresh Produce, Agricultural Research Organization, The 12 Volcani Center, Beit Dagan 50200, Israel

**Keywords:** transcriptome, proteome, *Trichothecium roseum*, antifungals, cuminal, postharvest

## Abstract

*Trichothecium roseum* is a harmful postharvest fungus causing serious damage, together with the secretion of insidious mycotoxins, on apples, melons, and other important fruits. Cuminal, a predominant component of *Cuminum cyminum* essential oil has proven to successfully inhibit the growth of *T. roseum in vitro* and in vivo. Electron microscopic observations revealed cuminal exposure impaired the fungal morphology and ultrastructure, particularly the plasmalemma. Transcriptome and proteome analysis was used to investigate the responses of *T. roseum* to exposure of cuminal. In total, 2825 differentially expressed transcripts (1516 up and 1309 down) and 225 differentially expressed proteins (90 up and 135 down) were determined. Overall, notable parts of these differentially expressed genes functionally belong to subcellular localities of the membrane system and cytosol, along with ribosomes, mitochondria and peroxisomes. According to the localization analysis and the biological annotation of these genes, carbohydrate and lipids metabolism, redox homeostasis, and asexual reproduction were among the most enriched gene ontology (GO) terms. Biological pathway enrichment analysis showed that lipids and amino acid degradation, ATP-binding cassette transporters, membrane reconstitution, mRNA surveillance pathway and peroxisome were elevated, whereas secondary metabolite biosynthesis, cell cycle, and glycolysis/gluconeogenesis were down regulated. Further integrated omics analysis showed that cuminal exposure first impaired the polarity of the cytoplasmic membrane and then triggered the reconstitution and dysfunction of fungal plasmalemma, resulting in handicapped nutrient procurement of the cells. Consequently, fungal cells showed starvation stress with limited carbohydrate metabolism, resulting a metabolic shift to catabolism of the cell’s own components in response to the stress. Additionally, these predicaments brought about oxidative stress, which, in collaboration with the starvation, damaged certain critical organelles such as mitochondria. Such degeneration, accompanied by energy deficiency, suppressed the biosynthesis of essential proteins and inhibited fungal growth.

## 1. Introduction

Postharvest fungal diseases are responsible for quality deterioration of fruits and vegetables and for resultant severe losses of produce during handling, transportation, and storage. Phytopathogen *Trichothecium roseum* is one of the most significant spoilage fungi in China, causing decay in stored apples [1], melons [2,3], grapes [4], strawberries [5], and oranges [6]. Presently the pathogen can be controlled predominantly by chemical fungicides [7]. Nonetheless, decline in the number of available fungicides to be used for postharvest treatment, reduced efficacy due to the emergence of pathogen resistance to these chemicals, and the growing public concerns over chemical residues in food and the environment have necessitated the seeking of alternative solutions to control the fungal spoilage and to guarantee food safety and consumer health [8]. 

Essential oils (EOs), generally recognized as safe by the US Food and Drug Administration [9], are natural substances with established antimicrobial activities [10,11] and are a candidate instrument for controlling postharvest disease [12,13]. Nonetheless, this candidacy is compromised by the expensiveness of these natural substances due to their generally low yields. However, exploring their mode of actions is valuable to the development of novel alternatives. 

The multicomponent EOs attribute their mode of action to various responsible components [14,15] rather than a simple and sole-mechanism action of an individual component. This wide mode action is an advantage of EOs over synthetic fungicides, but at the same time, it complicates the elucidation of the antifungal mechanisms of EOs. For this reason, the search for the antifungal mechanism of the active component of the EOs is of practical significance. Such pioneer research work on eugenol and cinnamaldehyde revealed their effect on critical enzymes, such as those involving fungal bioenergetics [16,17]. Nonetheless, the earlier biological events elicited by these EO components were not well understood. However, such an understanding might be of primary importance for the elucidation of the action modes of these compounds. 

Conventionally, the complex antifungal mechanisms of EOs were basically attributed to their lipophilicity. When interacting with cells, EOs can damage the cytoplasmic membrane by causing leakage of intracellular substances, thereby resulting in cell death [18]. Reports have indicated an effect of EOs on a decrease of energy acquisition following the sabotage of fungal mitochondria by enhanced accumulation of reactive oxygen species (ROS) [19,20]. Indeed, these findings are in line with some action modes of commercial antibiotics [21]. Accordingly, exploring novel targets could be an instrument for developing safer fungicides with fewer and mitigated side effects. 

The investigation of the interaction of fungal cells with antifungal EOs agents may provide a global analysis of cellular gene expression alteration upon treatment. Such analysis could give us opportunities to fully understand fungal responses to these compounds. Transcriptome sequencing and iTRAQ (isobaric tags for relative and absolute quantitation) proteomic sequencing can explore gene expression changes at transcript and translational levels, respectively, occasioned by different stimuli [22,23,24]. Given the post-transcript regulation, protein turnovers, and alternative translation rate, consolidation of transcriptome and proteome analyses, along with bioinformatics interpretation, could furnish us with a more thorough understanding of the gene networks involved [25] and of molecular events during the interaction of fungi with antifungals.

*Cuminum cyminum* EO and its predominant component cuminal (4-Isopropylbenzaldehyde) showed antifungal activity against *T. roseum* [26], providing us with a potential way to control fruit rot caused by this devastating pathogen. Nonetheless, their antifungal mechanisms have not yet been well investigated. In the present investigation, the antifungal activity of cuminal against *T. roseum* was assessed. RNA-seq transcriptome and iTRAQ proteome analysis were conducted on *T. roseum* upon cuminal exposure to elucidate the wide metabolic responses of the fungus to cuminal. Analysis of the fungal responses indicated that the treatment impaired the polarity of cytoplasmic membrane and triggered the reconstitution and dysfunction of the membrane, resulting in handicapped nutrients procurement and fungal starvation, therefore causing the inhibition of fungal growth. 

## 2. Materials and Methods

### 2.1. In Vitro Antifungal Activity Measurement

#### 2.1.1. Antigermination Test and Minimum Inhibitory Concentration (MIC) 

The modified method of Chitarra et al. [27] was used to evaluate the effect of cuminal on *T. roseum* conidia germination. *T. roseum* (Pers.: Fr.) Link was isolated from decayed muskmelon fruit with typical pink rot and preserved on potato dextrose agar (PDA) at 25 °C. The fungal conidia were washed from the surface of PDA plates with sterile 0.85% saline containing 0.1% Tween 80 (v/v) to prepare conidia inoculum and stored at 4 °C for further use. Cuminal (Sigma-Aldrich, Shanghai) was added to potato dextrose broth (PDB) by dissolving the requisite amounts of cuminal into PDB containing 0.05% (v/v) Tween 80 to prepare a series of concentrations (0, 0.05, 0.10, 0.15, 0.20, and 0.25 µL/mL), and a 0-concentrate one was used as the control. Slides inoculated with 20 µL conidia in the above PDB medium (10^6^ conidia/mL) were placed on moist filter paper in Petri plates, and the plates were parafilmed to avoid evaporation and incubated at 27 °C. When the germination rate of the control conidia reached 80, all of the corresponding treatment conidia on the slides were immediately fixed with lactophenol cotton blue to stop further germination and the germinated conidia of each slide were counted. The results were expressed as the antigermination rate using the formula:(1)$$\mathrm{GI}\left(\%\right)=[\left(G_{c}-G_{t}\right)/G_{c}]\times 100$$
where G_c_ and G_t_ represent the mean number of germinated conidia in the control and treated slides, respectively. Each treatment was performed in triplicate.

Minimum inhibitory concentration (MIC) of cuminal against *T. roseum* in PDB was determined by a serial dilution techniques [28] using 96-well microtitre plates. Cuminal was added in PDB with inoculum and the microplates were incubated at 27 °C for 72 h. The lowest concentration without visible growth under binocular microscope was defined as MIC.

#### 2.1.2. Antifungal Activity of Cuminal Vapor against Mycelial Growth 

In vitro antifungal assays of cuminal vapor were carried out using a modified method of Aguilar-Gonzaliez et al. [29]. Briefly, Mycelial disks (4-mm-diameter) from the periphery of 7-day-old cultures were centrally inoculated onto the PDA plates with the mycelium surface facing down. A piece of round Whatman No.1 filter paper (20-mm-diameter) was stuck to the inside center of each Petri dish lid and requisite amounts of cuminal were poured on each round paper to achieve concentrations of 0, 0.0625, 0.125, 0.25, 0.50, and 1.00 µL/mL relative to the air volume in the Petri dishes, and then plates were parafilmed to avoid vapor loss and incubated at 27 °C in the dark for 7 days. Among the plates, the 0-concentrate ones were used as control. The mean of two perpendicular diameters of the colony was measured. The rate of mycelial radial growth inhibition was calculated with formula:(2)MGI(%)=[(C−T)/C]×100
where C and T represent mycelial growth diameter in control and cuminal-added Petri plates, respectively. Three plates were used for each treatment as replications.

### 2.2. In Vivo Antifungal Activity

The in vivo antifungal activity of cuminal against *T. roseum* was tested on apple fruits using a modified method from the literature [28]. Apple fruits (cv. Fuji) were obtained from an orchard located in Jinning town, Gansu (China). Fruits free of wounds and rot, and as homogeneous in maturity and size as possible were selected. Fruits were surface-disinfected by immersion in a 2% sodium hypochlorite solution for 6 min, washed twice by immersion in sterile deionized water, and air-dried in shade to remove excess surface water. The fruits were wounded (3 mm deep and 3 mm wide) with a sterile nail at the equator, with two directly opposite wounds per fruit. Each treatment had two replicates with eight fruits per replicate. Cuminal was added to PDB containing 0.05% Tween 80 to reach cuminal concentrations of 0, 0.5MIC (2 μL/mL), and MIC (4 μL/mL), and *T. roseum* conidia were added to the above PDBs to reach a concentration of 10^6^ conidia/mL. The sets of conidia in PDBs were cultured in duplicate in a shaker (130 r/min) at 27 °C for 3 h. To investigate the sustaining efficacy of the cuminal treatment on *T. roseum* conidia, one copy of the above conidia set was centrifuged to remove cuminal-containing PDB and the harvested conidia were washed twice with distilled and sterile water by re-suspension and centrifugation. The washed conidia were re-suspended in fresh PDB to reach concentration of 10^6^ conidia/mL. Then, 20 μL of unwashed- and washed-conidia PDBs was placed into each wound, separately. The apple fruits were put into plastic boxes with sterile water to maintain a high relative humidity (95%) and kept at 25 °C for observations. The mean of perpendicular lesion diameters of each wound was measured on day 14. 

### 2.3. Electron Microscope Observations

One-hour-old PDB culture of *T. roseum* conidia was exposed to 4 µL/mL of cuminal while the control was without cuminal, and further cultured for 1 day. The fungal cells were harvested, and washed thrice with sterile 0.85% saline. The morphological and ultrastructural changes were observed under scanning electron microscopy (SEM) (Evo-18, Zeiss) and transmission electron microscopy (TEM) (Tecnai G2-T20 STwin 200kV, Fei Co.), respectively, after sample preparation was done according to the method of Dwivedy et al. [30]. Digital images were acquired with a charge-coupled device camera (Megaview III, Fei Company, Eindhoven, The Netherlands) using the microscope-attached software iTEM (Sift Imaging System, Münster, Germany).

### 2.4. Fungal Sample Preparation

The *T. roseum* conidia were incubated in 1000 mL Erlemeyer flasks containing 200 mL PDB inoculated with 100 μL conidia suspension (5 × 10^5^ conidia/mL), at 25 °C with shaking (200 r/min) for 96 h. Then, the flasks were supplied with fresh sterile PDB (200 mL) with 0.1% (v/v) Tween 80. Cuminal was consecutively added to certain flasks to reach a final concentration of 4 μL/mL (v/v). Flasks without cuminal addition were used as control. All flasks were cultured for a further 2 h under the same conditions as before. The mycelia were harvested by vacuum filtration on a sterile clean bench, washed thrice with sterile deionized water, and snap-frozen with liquid nitrogen. Three replicate cultures were performed for each treatment and control. All samples were stored at −80 °C until omics analysis. 

### 2.5. RNA Sequencing and Data Processing

#### 2.5.1. RNA Extraction and Sequencing Strategies

Total RNA was extracted with TRIzol reagent (Life Technologies, UK) from the prepared *T. roseum* mycelia samples. The libraries were sequenced on the Illumina HiSeq™2500 platform. Readings were analyzed by tool FASTQC, and the data of poor quality bases (phred ≥20), unexpected Illumina adapters and poly-A tails were removed using the Toolkit NGS QC v2.3.3 [31].

#### 2.5.2. Transcriptome Assembly and Annotations

De novo short read assembly was carried out with the software of Tophat and Cufflinks. The assembled readings were mapped to the complete genome of *Saccharomyces cerevisiae* (strain ATCC 204508 / S288c) (Baker’s yeast) using Tophat and Bowtie2. Unigenes were aligned to the NCBI NR Database, wherein the unigenes encoding proteins with high similarity (e < 1 × 10^−5^) to the known proteins were used for annotation. Gene ontology (GO) annotation was conducted with Blast2GO software. Kyoto Encyclopedia of Genes and Genomes (KEGG) annotation was performed with the database http://www.genome.jp/kegg/ pathway.html [32].

#### 2.5.3. Transcriptome Analysis

As described by Yan et al. [32], differential transcript accumulation in samples of controls and treatments was observed using bowtie2 and eXpress. The gene expression levels were calculated with fragments per kilobase per million method. Differentially expressed genes (DEGs) were judged based on Baggerley’s test with a significance level of <0.05 and the fold change >2 or <0.5. GO enrichment analysis of DEGs was performed by mapping them to GO terms in the database (Available online: http://www.geneontology.org/) and by calculating gene numbers for every term, and by hypergeometric testing to determine significantly enriched GO terms. KEGG pathway enrichment analysis was conducted on DEGs.

### 2.6. iTRAQ-Labeled Proteome Analysis

#### 2.6.1. Protein Extraction, Digestion and Labeling

Total mycelia proteins were extracted as described by Zhang et al. [23]. Briefly, the fungal samples were re-suspended in lysis buffer supplemented with protease inhibitor solution, and the suspensions were sonicated on ice for 15 min. After centrifugation at 25,000× *g* for 20 min at 4 °C, the sediments were discarded, and the expected proteins in supernatants were quantified and stored at −70 °C until digestion.

As described by Wisniewski et al. [33], proteins were digested using the filter-aided sample preparation method. iTRAQ labeling was performed using iTRAQ 8-plex reagent kits (AB Sciex, Framingham, MA, USA). iTRAQ reagent was rested at room temperature and centrifuged to the tube bottom, and isopropanol was added. iTRAQ reagent (100 μL) was transferred to the sample tubes and vortexed before spinning. The tubes were incubated at room temperature for 2 h and 200 µL of water was added to quench the labeling. The solutions were lyophilized and stored at −70 °C for analysis.

#### 2.6.2. Peptide Fractionation and Mass-Spectrometry Analysis

Peptide fractionation and mass-spectrometry analysis were routinely carried out as described by Zhang et al. [34], and the enzymatically hydrolyzed protein solution was re-suspended with 110 µL of eluent A (acetonitrile-H_2_O-formic acid, 2:98:0.1, v/v/v). Peptides were separated using an Agilent 1200 HPLC (Wilmington, DE, USA) with a guard column and a separation column, and UV detection wavelengths were 210 nm and 280 nm. Mass-spectrometry analysis was performed on a Triple TOF 5600 System (AB SCIEX, Foster City, CA, USA) fitted with a Nanospray III source (AB SCIEX, Framingham, US). Data was acquired under the following conditions: ion spray voltage 2.4 kV, 35 PSI curtain gas, 5 PSI nebulizer gas, and interface heater temperature 150 °C. The MS scans were acquired in 250ms, and each cycle time was fixed to 2.5s. The dynamic exclusion set was 22s. 

#### 2.6.3. Protein Identification and Quantification

The MS/MS spectra were processed using Protein Pilot Software v.5.0 (AB Sciex, Framingham, MA, USA) as described by Zhang et al. [34]. Laconically, only peptides with the 95% confidence interval were counted as the identified protein. Proteins with an average fold change >1.2 or <0.83 in treated groups compared to controls were regarded as differentially expressed proteins (DEPs). The experimental data from MS/MS were matched with the theory data to achieve protein identification with the following parameters: sample type, iTRAQ 8plex; cys alkylation, iodoacetamide; instrument, Orbi MS (sub-ppm) and Orbi MS/MS; and the search effort was thorough.

#### 2.6.4. Bioinformatics Analysis of DEPs

Homology mapping of the identified proteins to *S. cerevisiae* (ATCC 204508/S288c) was conducted based on sequence similarity, and their biological function information analysis was carried out through comparative analysis. An integrated GO enrichment and KEGG pathway analysis of DEPs was performed with OmicsBean (Available online: http://www.omicsbean.cn). The GO term or pathway enrichment of DEPs were considered as significant when *p* < 0.05 [35]. To examine possible post-transcriptional regulation and to achieve in-depth understanding of the proteome and transcriptome data, correlation analysis was conducted.

### 2.7. Protein–Protein Interaction Analysis

For DEPs and the proteins encoded by DEGs involved in the specific KEEG pathways, their predicted protein–protein interactions (PPI) were analyzed from the STRING database [36]. All these interactions were visualized using the Cytoscape tool.

### 2.8. Statistical Analysis

All antifungal tests were conducted in triplicate; omics analysis was with 3 biological replicates. Results were statistically processed and subjected to analysis of variance. Means significantly different were separated by the Tukey test using SPSS version 2019.

## 3. Results

### 3.1. Antifungal Activity in Vitro

In PDB medium containing various concentrations of cuminal (0.05 to 0.25 µL/mL), the inhibition of the conidial germination was observed when the effect was in a dose dependent manner (Figure 1A). In the bioassay matrix, the cuminal obtained an inhibition rate of 30.2% at the lowest concentration 0.05 µL/mL. When the concentration elevated, the antigermination efficacy was increased. However, at relatively higher concentrations, the ascending of the concentration showed no significant boost of antigermination rate. A maximum rate of 96.1% was observed at 0.25 µL/mL of cuminal.

Via a modified microdilution technique, the MIC of cuminal against *T. roseum* in PDB medium was established to be 4 µL/mL. This data was used in the implementation of in vivo test and fungal sample preparation for the following omics analysis. 

By virtue of the practical value of the essential oil fumigation, the inhibition effect of cuminal vapor against *T. roseum* mycelia was tested. Considering the effective vapor concentrations were not in proportion to the added cuminal doses, an equal-ratio gradient concentration set was used in the measurement of the vapor contact inhibition of mycelial growth. Under this bioassay matrix, the elevation of cuminal concentrations obtained stable increase of the mycelial growth inhibition (Figure 1B), where the lowest concentration (0.0625 µL/mL) showed an inhibition of 12.6%, and the concentration of 0.5 µL/mL reached an inhibition of 99.3%, with no significant increase when the concentration was further doubled.

### 3.2. In Vivo Apple Fruit Assay

The results on the artificially inoculated apple fruits with cuminal-treated *T. roseum* cells showed a noticeable suppression of the fungal growth by this agent in the host matrix (Figure 2). The lesion diameters in the treated apple fruits were significantly shorter (*p* < 0.05) than that of the control sets. The in vivo assay showed that both cuminal treatments at 0.5MIC and MIC well demonstrated an antifungal effect on apple fruits. Notably, the in vivo assay disclosed the sustained inhibition of the growth of *T. roseum* by cuminal treatment: after removing the surplus cuminal by washing with the saline, the treatment cells were still devitalized on the apple fruits. This was an interesting finding; the in vivo assay, unlike most other similar in vivo assays, indicated that cuminal treatment might disorder the fungal cells and that removing the remaining cuminal in the matrix cannot revive their original vitality.

### 3.3. Effects on Conidial Morphology and Ultrastructure

In SEM analysis, the conidia of *T. roseum* exposed to cuminal were found inflated and distorted in comparison to the control set (Figure 3A,B). Obvious roughness was seen on the conidial surface, indicating that the relationship between cell wall and plasmalemma was changed upon cuminal exposure, which denotes the membrane as the site of action. Upon exposure to cuminal, conspicuous deformities of cellular inner structures and organization were observed in TEM (Figure 3C,D). In treatment cells, detachment of the plasmalemma from the cell wall occurred with formation of small lomasomes and decrease in the cell matrix, possibly due to the decreased thickness of plasmalemma tentatively caused by the reconstitution of the plasmalemma. Another contingent event of the decreasing of plasmalemma thickness was the reduction of the cell matrix on account of the leakage of cellular components. Moreover, manifest decomposition of the cellular organelles and decline of the inner organization were observed in line with the debilitated cell vigor, shown both in in vitro and in vivo tests. Notably, abnormal cytoplasm coagulation indicated by high electron density was imaged, suggesting an unconquerable oxidative stress suffered in the treatment cells.

### 3.4. Fungal Transcriptome Profiles upon Cuminal Exposure 

In total, 14897 unigenes were assembled after de novo transcript splicing (Appendix A), and among them, 2825 genes were differentially expressed genes. Cuminal exposure up regulated 1516 genes and down regulated 1309 genes (Figure 4A,B). For these DEGs, 248 genes were highly up regulated (>10-fold change) at the transcriptional level, including 112 genes that were almost not transcripted in the control samples (the fold change values were marked as ‘inf’ in the Appendix A). Meanwhile, 381 genes were deeply down regulated with fold-changes <0.1, including 78 genes barely even expressed in the cuminal treated samples at transcriptional level (Appendix A). 

### 3.5. GO Analysis of DEGs

The DEGs were divided into up and down subgroups. Most of the DEGs were annotated with GO terms according to their functions, and these terms were mapped into “biological processes”, “cellular components”, and “molecular functions” categories. For up regulated DEGs, the terms most associated were metabolic process, cellular process, cell, cell part, and catalytic activity. Most enriched GO terms of up DEGs were single-organism process (315, 20.78%), single-organism cellular process (290, 19.13%), single-organism metabolic process (219, 14.45%), and catabolic process (102, 6.73%) in the biological process category. Additionally, genes involved in oxacid metabolism (70, 4.62%) and oxi-reduction process (67, 4.42%) were also up regulated. High enrichment in the subcellular components localized in the cell (342, 22.56%), cell part (341, 22.49%), intracellular (320, 21.11%), and intracellular part (315, 2.078%). Also, the mitochondrion (73, 4.82%), mitochondrial part (54, 3.56%), and mitochondrial matrix (26, 1.72%) were enriched with a noticeable amount of up regulated DEGs. In the molecular function category, catalytic activity (228, 15.04) and ion binding (166, 10.95%) were the most modulated entries involving the up regulated DEGs. Besides ion binding, a large part of the up regulated DEGs were also involved in the binding related function of other molecular groups including cofactors and other small molecules (Appendix A). 

The situation of GO analysis of down DEGs was different from that of up DEGs. Although most of the down DEGs were still involved in the biological process of metabolic process (262, 20.02%) and single-organism cellular process (236, 18.03%), several key processes including oxidation-reduction process (50, 3.82%), negative regulation of cell cycle (20, 1.53%), post replication repair (8, 0.6%), negative regulation of cell cycle process (15, 1.15%), and metabolic process relating to nucleotide and nucleoside derivatives were also enriched with a noticeable quantity of down regulated DEGs, and these processes are crucial to fungal growth. Most of the down regulated DEGs belong to cellular components of cell part (277, 21.16%), cell (277, 21.16%), intracellular part (258, 19.71%), intracellular (259, 19.77%), and cytoplasm (212, 16.20%), but replication fork, protein–DNA complex, ribo–nucleoprotein granule, septin cytoskeleton, and mini-chromosome maintenance protein complex were the related cellular components of a noticeable number of down regulated DEGs too. As to the molecular functions of the down DEGs, most of them were mapped into catalytic function and binding with other molecules, including entries of nucleotide and nucleoside derivatives binding (Appendix A). 

### 3.6. KEGG Metabolic Pathways of DEGs

To investigate the enrichment situation of the DEGs in metabolic pathways, these genes were mapped into the KEGG pathways with the Blast2GO program. In total, the 2825 DEGs were mapped into 216 metabolic pathways in which biosynthesis of amino acids (202, 7.15%), carbon metabolism (180, 6.37%), and ribosome (164, 5.81%) were the top three ones. For clearer understanding the effects of cuminal exposure on the fungal metabolism, the KEGG enrichment of DEGs was analyzed for up and down regulated DEGs, separately (Figure 5). A noticeable number of up regulated genes enriched the metabolic pathways (89%), biosynthesis of secondary metabolites (45%), biosynthesis of antibiotics (41%), carbon metabolism (19%), and biosynthesis of amino acids (18%). In particular, citrate cycle, pyruvate metabolism, glycerophospholipid metabolism, mRNA surveillance pathway and peroxisome were also enriched with notable DEGs. From an upper hierarchical view, the top enriched KEGG ways were global and overview maps (A0), carbohydrate metabolism (AA) and other unknowns (HA) (Figure 5A). For down regulated DEGs, the top enriched KEEG ways were metabolic pathways (84%), biosynthesis of secondary metabolites (37%), antibiotics biosynthesis (34%), cell cycle (17%), and glycolysis/gluconeogenesis (12%). From an upper hierarchy, most of these down regulated DEGs function in global and overview maps (A0), carbohydrate metabolism (AA), cell growth and death (DC), and other unknowns (HA) (Figure 5B). 

To evaluate the effects of cuminal exposure on the different KEGG pathways of *T. roseum*, functional enrichment analyses, where an index (rich factor) can evaluate how severely the treatment interferes in the pathways, were conducted (Figure 6). The rich factor indicates the ratio of the number of DEGs in a pathway to the number of genes annotated in the pathway. Higher rich factors define greater degrees of enrichment. ABC (ATP-binding cassette) transporters, lipid metabolism, valine, leucine and isoleucine metabolism, steroid biosynthesis, tryptophan metabolism, and propanoate metabolism had greater enrichment of up regulated DEGs. Additionally, certain large pathways such as metabolic pathways embraced a larger number of up regulated DEGs, but their enrichment factors were not high. Certain key pathways such as citrate cycle, pyruvate metabolism, and glycerophospholipid metabolism have both considerable rich factors and DEG numbers (Figure 6A).

The situation of functional enrichment of down regulated DEGs was different from that of up regulated DEGs. Concisely, the advanced glycation end-product and receptor for advanced glycation end-product signaling pathway and hippo signaling pathway were with higher in rich factors. Tyrosine metabolism, starch and sucrose metabolism, fatty acid degradation, DNA replication, and glycolysis/gluconeogenesis were the second enriched group of pathways, both with considerable rich factors and DEG numbers (Figure 6B).

### 3.7. Global Proteome Changes in Response to Cuminal Exposure

Using the Mascot search engine, the iTRAQ LC-MS/MS investigation generated 94463 spectra and identified 1485 proteins, in which 225 were differentially expressed (Appendix A). Among these DEPs, 90 were up regulated while 135 were down regulated (Figure 7).

As regards the magnitude of regulation, 21 proteins were highly up regulated with a fold-change >2, and 11 proteins were extremely low expressed with a fold-change <0.5. Of these DEPs, phosphatidylserine decarboxylase proenzyme 1 (mitochondrial) (*PSD1*) was the most highly expressed, with an increase of more than 17-fold and aspartate-tRNA ligase (cytoplasmic) (*DPS1*) increased by nearly 10-fold. In contrast, checkpoint serine/threonine-protein kinase (*BUB1*), an enzyme involved in cell cycle checkpoint enforcement, was the lowest expressed protein, decreasing by nearly 7-fold.

### 3.8. GO Annotation Analysis of the DEPs

Of the 225 DEPs, 212 were annotated in GO terms classified into biological process, cellular components and molecular function. The prominently enriched GO terms are demonstrated in Appendix A. Most of the 90 up regulated DEPs were mainly involved in small molecular metabolic process (15, 16.67%), carboxylic acid metabolic process (10, 11.11%), oxoacid metabolic process (10, 11.11%), organic acid metabolic process (10, 11.11%), small molecule biosynthetic process (7, 7.78%) and cellular amino acid metabolic process (6, 6.67%). Certain other up regulated DEPs enriched the metabolism of carbohydrate and amino acid derivatives. In terms of cellular components, the up regulated DEPs were mainly localized in cell part (30, 33.33%), cytoplasm (26, 28.89%), mitochondrial matrix (4, 4.44%), mitochondrial protein matrix (3, 3.33%), mitochondrial nucleoid (2, 2.22%) and some mitochondrial complexes. Regarding the molecular function enrichment of these up regulated DEPs, the most predominant enrichment was catalytic activity (27, 30%), followed by carboxylic ester hydrolase activity (3, 3.33%), transferring nitrogenous activity (2, 2.22%), and transaminase activity (2, 2.22%). Other molecular functions of up regulated DEPs were diverse and beyond the list (Appendix A). For biological processes, the 135 down regulated DEPs mainly enriched the organonitrogen compound metabolic process (26, 19.26%), cellular biosynthetic process (25, 18.52%), organic substance biosynthetic process (25, 18.52%) and organonitrogen compound biosynthetic process (20, 14.82%). Other biological processes enriched with the down regulated DEPs included translation, ribosome biogenesis and rRNA processing. These down regulated DEPs noticeably localized in the cellular components of intracellular (32, 23.70%), intracellular part (32, 23.70%), cytoplasm (29, 21.48%), cytoplasm part (25, 18.52%), intracellular organelle part (24, 17.78%), macromolecular complex (22, 16.30%), intracellular non-membrane-bounded organelle (20, 14.81%) and non-membrane-bounded organelle (20, 14.81%). Additionally, ribosome related components were also well enriched with down regulated DEPs. The molecular functions of these down regulated DEPs included the structural constituent of ribosome (15, 11.11%), structural molecule activity (15, 11.11%), RNA binding (10, 7.41%) and others such as transferase activity (Appendix A).

### 3.9. KEGG Pathway Analysis of DEPs

The Blast2GO KEGG enrichment analysis showed that the most significantly enriched pathways of some of the 90 up regulated DEPs were metabolic pathways (13%), biosynthesis of secondary metabolites (10%) and biosynthesis of antibiotics (8%). Particularly, carbon metabolism, citrate cycle and fatty acid degradation were well enriched with certain up regulated DEPs (Figure 8A).

For down regulated DEPs, their pathway enrichment was mainly in ribosome (15%), ketone metabolism and taurine metabolism (Figure 8B). When evaluated with the rich factor, the enrichment of the above DEPs exerted a different extent of influence on their corresponding pathways. With higher rich factors, the enrichment of up regulated DEPs probably and positively modulated alpha-linolenic acid metabolism, tyrosine metabolism and fatty acid degradation. Meanwhile, biosynthesis of secondary metabolites, biosynthesis of antibiotics and citrate cycle were enriched with more up regulated DEPs but with lower rich factors (Figure 9A). In contrast, the situation of down regulated DEPs was different, in that their enrichment conferred higher rich factors for minor pathways such as taurine and hypotaurine metabolism, and synthesis and degradation of ketone bodies, but a lower rich factor for a larger pathway such as ribosome, with 15 down regulated DEPs being enriched (Figure 9B).

### 3.10. Integration Analysis of Transcriptome and Proteome 

#### 3.10.1. Correlations of Proteome with Transcriptome

All three biological replicates demonstrated similar replications in the gene expression at both mRNA (Appendix A) and protein (Appendix A) levels. Among the 1485 identified proteins of the proteome, 1362 ones were annotated. The correlation number of these annotated proteins with annotated unigenes in transcriptome was 1305 (95.82%) (Appendix A) and these correlations are graphically shown in Appendix A. Given the larger number of DEGs, we carried out a one-by-one search for the corresponding gene in transcriptome to each DEP. All DEPs found their encoding genes based on the matching of their IDs (Appendix A), but some of these genes were not DEGs. Heat map analysis of correlations between certain DEPs and their encoding genes in DEGs was performed (Appendix A). Furthermore, among annotated DEPs, 26 up and 24 down regulated DEPs had their encoding DEGs with the same changing tendency, and others were not (Appendix A). 

Noticeable up/up and down/down genes were focused on mitochondria and ribosomes. They function in diverse types of metabolism including citrate cycle, stress response, phospholipid metabolism, urea cycle, transcription, fatty acid β-oxidation, electron transport, cellular protein catabolic process, regulation of cell cycle, antibiotic resistance, glucosidic compound degradation, energy metabolism, translation, redox homeostasis and some cellular components biogenesis. Interestingly, up regulated genes more enriched the stress response, citrate cycle, phospholipid degradation and other catabolism, while considerable genes involved in glycolysis, cell cycle and anabolism were generally down regulated. 

#### 3.10.2. Pathway Integration of DEGs and DEPs 

The DEPs were involved in 86 KEGG pathways while the DEGs were in more than 163 KEGG pathways. To analyze the shared pathways of DEGs and DEPs, the correlation analysis of top enriched KEGG pathways of DEPs and DEGs were conducted. The most correlated KEGG pathways were metabolic pathways (22), biosynthesis of secondary metabolites (13), biosynthesis of antibiotics (12), carbon metabolism (6) and biosynthesis of amino acids (5) (Appendix A).

Among these shared pathways, the biosynthesis of antibiotics is of interest. Moreover, ribosome, oxidative phosphorylation, terpenoid backbone biosynthesis, fatty acid metabolism, 2-oxocarboxylic acid metabolism, glycolysis/gluoconeogenesis, citrate cycle, steroid metabolism, fatty acid degradation, pantothenate and CoA biosynthesis, and valine, leucine and isoleucine degradation were well enriched during the fungal response to cuminal exposure. Because of the large number of genes involved in these pathways and the multifunctional roles of these pathways, these specific pathways might be of critical importance in demonstrating the antifungal mechanism of cuminal against *T. roseum*. To further understand the importance of these KEGG pathways during the response to cuminal, the correlation of the KEGG pathways enriched in transcriptome and proteome was analyzed (Figure 10). Besides certain basic metabolic pathways, KEGG pathways correlated in both transcriptome and proteome significantly embraced valine, leucine and isoleucine degradation, biosynthesis of antibiotics, glycerolipid metabolism, fatty acid degradation, propanoate metabolis, and glycolysis/gluconeogenesis. Meanwhile, cell cycle, citrate cycle, glycerophospholipid metabolism, cysteine and methionine metabolism, pentose and glucuronate interconversions, and peroxisome, pyruvate metabolism and steroid biosynthesis were only enriched in transcriptome. Terpenoid backbone biosynthesis, synthesis and degradation of ketone bodies, ribosome, and 2-oxocarboxylic acid metabolism were only enriched in the proteome. These differences of pathway enrichment suggest that post transcription regulation plays an important role in the gene expression. 

#### 3.10.3. PPI upon Cuminal Exposure

KEGG pathway analysis explained how an individual protein works in the context of a large network of various related proteins. Based on GO and KEGG analyses of DEGs and DEPs, proteins involved in central carbon metabolism and energy acquisition, membrane composition and drug effluxing, stress mediation and antioxidative defenses, ribosome formation and protein biosynthesis, and cell cycle and multiplication were further investigated for their PPIs. These interactions are of critical importance for the specific KEGG pathways underpinning the above metabolic blocks and were established based on the STRING database (Figure 11).

Considering the lipophilicity of cuminal, the plasmalemma lipid bilayer might first be affected by the cuminal invasion. We think this invasion was the first biological event among the interactions of the molecule and the fungus; this invasion might affect the constitution and functions of the plasmalemma. To manifest how the cuminal intrusion impacts the constitution and functions of plasmalemma, the protein–protein interaction network of proteins encoded by DEGs functioning in the fungal membrane composition and drug effluxing was established (Appendix A). Taking into consideration that central carbon and energy metabolism is of pivotal importance in the vitality and growth of any biospecies, the protein–protein interaction network of proteins encoded by DEGs functioning in the fungal carbohydrate and energy metabolism was shown (Appendix A). 

### 3.11. T. roseum Responses to Cuminal Stress

By a detailed analysis of specific functions of DEGs and DEPs involved in metabolic blocks mentioned in PPI analysis, the responses of *T. roseum* to cuminal exposure were further explored. In GO analysis of the DEGs and DEPs, the most enriched biological processes were catabolic process, oxidation-reduction process, and single-organism process; a large part of these DEGs and DEPs belong to cell part, mitochondria, membrane-bounded organelles and cytoplasm. Moreover, KEEG pathway enrichments showed that central carbon metabolism, stress responses, membrane components degradation and ribosomes were the most regulated pathways by cuminal exposure. Accordingly, how the DEGs or DEPs involved in these metabolic blocks correlated and how their regulation was orchestrated were further elaborated to uncover the system response of the fungus to cuminal.

#### 3.11.1. Central Carbon Metabolism and Energy Acquisition

Central carbon metabolism is one of the most basic metabolic pathways in all living cellular organisms and is considered to be the underpinning and driving metabolism affecting cell physiology and vitality [37]. Environmental conditions can modulate the metabolic fluxes of branches of central carbon metabolism, helping the organisms negotiate the impacts of adverse conditions [38]. Cuminal exposure regulated the gene expression of *T. roseum* to adjust the fuel flux distribution among the branches of central carbon metabolism. First, glycolysis of the fungus slowed overall when *HXK1* (Hexokinase-1), *PGK1* (Phosphoglycerate kinase), and *CDC19* (Pyruvate kinase 1) were down transcripted, and the *PFK2* (6-phosphofructokinase subunit beta) was up transcripted. Consistently, the HXK1, a speed-limit key enzyme catalyzing the conversion of glucose to glucose-6-phosphate during glycolysis was down regulated at protein level. 

The overall situation of gluconeogenesis was different from that of the glycolysis, wherein six genes (*ALD5*, *ACS2*, *THI3*, *ADH6*, *ACS1*, and *LAT1*) were up regulated at transcriptional level and one gene (*ADH5*) was up regulated at the protein level. *PCK1* (phosphoenolpyruvate carboxykinase), a critical enzyme in gluconeogenesis, was severely down regulated in both transcriptome and proteome, which resulted in oxaloacetate from the upper reactions of gluconeogenesis not being efficiently converted to phosphoenol–pyruvate and thereby having to be metabolized through the citrate cycle. Interestingly, *ACS1* and *PDC6* encoding isoenzymes of ACS2 and THI3, respectively, were down regulated, implying fine but unraveled regulation of the gluconeogenesis. For the pentose phosphate pathway, no DEGs or DEPs enriched its oxidative phase, but in the non-oxidative phase, *PFK2* catalyzing the one-way reaction of β-D-fructose-6P to β-D-fructose-1,6P2 was up transcripted, thereby supplying the non-oxidative phase with the D-glyceraldehyde-3P. Moreover, in the non-oxidative phase, two genes (*PGM2* and *TAL1*) and one gene *TKL1* were down regulated at transcriptional and translational level, respectively. Nonetheless, owing to the supplement of metabolites to the non-oxidative phase from other linked metabolisms, such as pentose and glucuronate interconversions, and other carbohydarate metabolisms, the effects of the down regulation of certain non-oxidative phase enzymes on the generation of reducing power from pentose phosphate pathway might be compensated for somewhat, so the cells still possessed weak anti-oxidative capacity.

Cuminal exposure regulated the citrate cycle more positively. A total of seven genes (*KGD1*, *LSC2*, *IDP3*, *SDH1*, *CIT2*, *LAT1*, and *IDP1*) were up regulated at the transcript level, and among them, *CIT2*, *IDP1* and *KGD1* encode the key enzymes of the cycle citrate synthase, isocitrate dehydrogenase, and 2-oxoglutarate dehydrogenase, respectively. However, *LSC1*, *CIT1* and *MDH2* were down regulated. The down regulation of *LSC1* and *CIT1* might be offset by the significant overexpression of their corresponding family gene *LSC2* and *CIT2*. Although the *MDH2* was down regulated, its isoenzyme MDH1 was overexpressed in proteome. Interestingly, one critical enzyme of the citrate cycle, the 2-ketoglutarate dehydrogenase (KGD1) was highly (2.68-fold) overexpressed in proteome too. Furthermore, no down regulated proteins enriched the citrate cycle. Overall, metabolism of the citrate cycle was enhanced largely due to the elevation of β-oxidation, val, leu and Ile degradation, and other amino acid degradation and these pathways were significantly enriched in both omic datasets.

The regulation of β-oxidation seems more straight and positive than other metabolisms. The enzyme 3-ketoacyl-CoA thiolase (*POT1*) was up regulated at both mRNA (5.52-fold) and protein (1.57-fold) levels, and this enzyme catalyzes the releasing of acetyl-CoA from the β-oxidation and has a key role in fatty acid degradation [39]. Long-chain acyl-CoA synthetase (*FAA1*), an enzyme of the ligase family, was extremely (373.42-fold) up regulated at mRNA level and the enzyme activates the breakdown of complex fatty acids and is also involved in the esterification, with concomitant transport, of exogenous long-chain fatty acids into metabolically active CoA thioesters for subsequent degradation. Additionally, two genes (*DIT2* and *ALD5*) functioning in affiliated metabolism of the β-oxidation were up regulated too, and they degrade alkanes, 1-alcohols and aldehydes before β-oxidation. No down regulated DEGs enriched the β-oxidation pathway. Furthermore, alcohol dehydrogenase (ADH5) was overexpressed at the protein level, enhancing the transferring of 1-alcohols to aldehydes and preparing more fuels to β-oxidation. Altogether, the elevated β-oxidation supplied more acetyl-CoA molecules to citrate cycle, and thus the fungus obtained a compensation for the down regulation of glycolysis. Given the just mentioned situations, the productivity of the glycolysis, even down regulated in general, may supply some intermediates to the pentose phosphate pathway to ensure subsistence reducing power during the fighting of stress. 

In eukaryotes, oxidative phosphorylation is carried out by a series of protein complexes composing the electron transport chains within the inner membrane of mitochondria. Cuminal exposure did not regulate the genes encoding the subunit of complex I (NADH-coenzyme Q oxidoreductase). Of complex II (Succinate-Q oxidoreductase), succinate dehydrogenase (ubiquinone) flavoprotein subunit 1 (*SDH1*) and subunit 2 (*YJL045W*) were down regulated at mRNA level. For complex III (Q-cytochrome c oxidoreductase), ubiquinol–cytochrome c reductase core subunit 2 (*QCR2*) was down transcripted. Consistently, of complex IV (Cytochrome c oxidase), cytochrome c oxidase assembly protein subunit 11 (*COX11*) was highly down regulated at mRNA level, and cytochrome c oxidase subunit 4 (COX5A) was down at the protein level. More importantly, mitochondrial ATP synthase subunit 9 (*OLI1*) of complex V (ATP synthase) was down regulated, though the β-subunit of the F-type ATPase (*ATP2*) was up regulated. Supply of phosphate (Pi) from PPi hydrolysis is necessary for the well performance of complex V, but the gene of inorganic pyrophosphatase, mitochondrial (*PPA2*) was down regulated at the mRNA level, which may weaken the complex V.

#### 3.11.2. Membrane Constitution and Drug Effluxing 

Plasmalemma plays a pivotal role in maintaining the sound function of most other cellular components. Membrane component reconstitution could adapt the cells to the environmental changes. Glycerophospholipids compose the framework of the plasmalemma and their metabolism essentially is a restructuring of the cellular membranes [40]. Cuminal exposure up regulated the transcription of certain genes functionally involved in membrane lipids metabolism. Among the proteins they code, choline kinase (*CKI1*) transfers the choline into phosphor–choline, and this activated choline molecule is a precursor of phosphatidylcholine. CDP-diacylglycerol-inositol 3-phosphatidyltransferase (*PIS1*) catalyzes CDP-diacylglycerol to phosphatidyl-1D-myo-inositol involving in inositol phosphate metabolism and GPI-anchor biosynthesis. 1-Acylglycerone phosphate reductase (*AYR1*) and lysophosphatidate acyltransferase (*SLC1*) catalyze successive bio-reactions to transfer 1-acyl-glycerone to 1, 2-diacyl-sn-glycerol-3P. Diacylglycerol kinase (*DGK1*) converts the 1, 2-diacyl-sn-glycerol to 1, 2-diacyl-sn-glycerol-3P that then participates in the inositol phosphate metabolism or is further converted to phosphatidylethanolamine. In fungal cells, lysophospholipid acyltransferase (*ALE1*) catalyzes the 1-acyl-sn-glycero-3-phosphoethanolamine to phosphatidylethanolamine, and ethanolaminephosphotransferase (*EPT1*) catalyzes the integration of CDP-ethanolamine and 1, 2-diacyl-sn-glycerol to form phosphatidylethanolamine. The phosphatidylethanolamine N-methyltransferase (*CHO2*) converts phosphatidylethanolamine to monomethyl–phosphatidylethanolamine, an intermediate to synthesize the phosphatidylcholine. Altogether, the up regulated glycerophospholipid metabolism genes could elevate the biosynthesis of phosphatidylcholine and inositol phosphate metabolism. The down regulation of cardiolipin synthase (*CRD1*) may limit the conversion of CDP-diacyl-glycerol to cardiolipin and thus more CDP-diacyl-glycerol molecules could be metabolized into phosphatidylethanolamine. Concordantly, in the proteome, phosphatidylserine decarboxylase (PSD2) that catalyzes phosphatidyl-L-serine to phosphatidylethanolamine was up regulated. In contrast, no down regulated DEPs enriched the glycerophospholipid metabolism.

Sterols are hallmarks of the eukaryotic membranes and reconstitution of these components modulates the function of the membranes. Four genes (*ERG25*, *ERG9*, *ERG6*, and *ERG5*) involved in sterol biosynthesis were down regulated. De novo biosynthesis of steroids derived its precursors from terpenoid backbone biosynthesis. Farnesyl-diphosphate farnesyltransferase (*ERG9*) catalyzes the producing of squalene, and highly down regulation of the enzyme suggests less squalene generated, which therefore limits the biosynthesis of the subsequent intermediates of the pathway. The down regulation of methylsterol monooxygenase (*ERG25*) may limit the biosynthesis of zymosterol, an important precursor of ergosterol biosynthesis, subsequently catalyzed by sterol 24-C-methyltransferase (*ERG6*) and sterol 22-desaturase (*ERG5*), both of which were down transcripted, and ergosterol is the precursor of vitamin D2. Although certain genes coding enzymes catalyze the downstream metabolism since ergosterol were up regulated at transcript level, this up regulation does not necessarily mean an elevation of the sterol biosynthesis, but rather, a more sophisticated and unraveled regulation of steroids metabolism under stress. 

Facing stresses, microbial cells could re-modulate the ratio of saturated to unsaturated fatty acids to survive the adverse conditions [41]. Cuminal exposure down regulated two genes coding acyl-CoA oxidase (*POX1*) and elongation of fatty acids protein 2 (*ELO2*) at mRNA level, both of which are involved in the biosynthesis of polyunsaturated fatty acids especially linolenic acid, arachidonic acid, icosapentaenoic acid and docosahexaenoic acid. Nonetheless, 3-ketoacyl-CoA thiolase (*POT1*, 5.52-fold in transcriptome; 1.57-fold in proteome) was up regulated in both omic datasets, but the enzyme elevation herein strengthens the β-oxidation. These adjustments of the relevant gene expression suggested that the unsaturated fatty acids biosynthesis was limited and that breakdown of lipid fatty acids via β-oxidation increased. The gene (*FAA1*) coding the long-chain-fatty-acid-CoA ligase 1, an important key enzyme of fatty acid degradation, was highly overexpressed (373.42-fold) at mRNA level. Another two genes, *DIT2* and *ALD,* encoded enzymes involved in the degradation of long chain alcohols and aldehydes were up regulated too. Additionally, up regulation of alcohol dehydrogenase 5 (ADH5) at the protein level could also facilitate the catabolism of alcohols from lipid degradation.

Cuminal treatment up regulated four genes (*PDR5*, *ATM1*, *SNQ2* and *STE6*) relating to drug efflux pumps functions. Among the proteins coded by these four genes, pleiotropic ABC efflux transporter of multiple drugs (*PDR5*) is responsible for active effluxing of weakly charged organic compounds, and confers resistance to numerous chemicals including antiseptics, antibiotics, herbicides, mycotoxins and other antifungals [42]. Consistently, the PDR5 was overexpressed (4.51-fold) at protein level. Iron–sulfur clusters transporter ATM1, in concert with glutathione, functions in the export of a substrate required for cytosolic-nuclear iron–sulfur protein biogenesis and cellular iron regulation, protects the cell against oxidative stress, and plays a role in vacuolar functions in isolating and exporting materials that might be harmful or a threat to the cell [43]. ABC export permease SNQ2 acts as pleiotropic drug resistance transporter and confers cells the capability of exporting drugs [44]. Alpha-factor-transporting ATPase (STE6) is a full-size ABC-B transporter distributed in all organisms and is involved in multidrug resistance [45].

#### 3.11.3. Stress Mediation and Antioxidative Defenses

Environmental toxic agents can induce the increase of ROS level [46], resulting in oxidative stress (OS) if the ROS are over accumulated in the organisms. For survival upon cuminal exposure, the fungus drew on antioxidant arsenals to minimize the detrimental effects of oxidative stress, and several genes involved in the peroxisome pathway were modulated. Peroxisomal catalase A (*CTA1*), occurring in almost all aerobic organisms and being able to protect cells from the toxic effects of hydrogen peroxide during the response to reactive oxygen species [47], was overexpressed. Two genes (*IDP1* and *IDP3*) encoding isocitrate dehydrogenase [NADP] (IDH) were also overexpressed. IDH is involved in the NADPH regeneration and hence cellular processes depending on NADPH, and IDH suppresses intracellular and mitochondrial ROS level [48]. Highly (39.48-fold) expressed *SYM1* encodes protein Mpv17l involved in cellular response to stress and required to maintain mitochondrial DNA integrity [49]. Mpv17l is also involved in peroxisomal reactive oxygen species metabolism and protects cells against mitochondrial oxidative stress by activation of Omi/HtrA2 protease [50]. *SOD2* encoding superoxide dismutase [Mn] of mitochondria was highly transcripted (33.28-fold), and this protein scavenges toxic superoxide anion radicals normally produced within the cells [51]. 

Cuminal exposure positively regulated several genes involved in mitophagy. Mitophagy, a selective degradation of mitochondria by autophagy, belongs to the oxidative defenses and promoters of life extension when correctly regulated [52]. Up transcripted cell wall integrity and stress response component 3 (*WSC3*, 2.26-fold) is a cell surface stress sensor detecting mitophagy-inducing stimuli and activating the Hog1 and Slt2 signaling pathways, facilitates the formation of the autophagosome surrounding the mitochondria and its eventual fusion with vacuoles for mitochondrial degradation [53], and also elevates the pleiotropic drug resistance [54]. With an ATP-independent isopeptidase activity, ubiquitin carboxyl-terminal hydrolase 3 (*UBP3*, 5.64-fold) was up transcripted, and the enzymeis required for efficient stress granule assembly in *Saccharomyces cerevisiae* [55]. Stress granules have long been proposed to function in protecting RNAs from stress conditions [56]. Up transcripted *SLG1* plays a role in regulating the entering or exiting the cell cycle and has a functional link between mitochondrial RNA editing and responses to abiotic stress [57]. Casein kinase II subunit alpha (*CKA1*) was up regulated at mRNA level and is an enhancing factor in abiotic stress signaling via modulating the expression of some molecular elements in retrograde signaling [58]. Transcriptically up regulated UBP3-associated protein BRE5 (*BRE5*) is involved in the formation of stress granules appearing when cells are under stress [59]. 

Consistently, down regulation of the following genes demonstrates that the fungal cells experienced an increased stress and responded to it as a price to survival. Catalytic mitochondrial inner membrane i-AAA protease super-complex subunit YME1 (*YME1*) is required for mitochondrial inner membrane protein turnover and is important to maintain the integrity of the mitochondrial compartment. At the protein level, this protein was up regulated 1.57-fold too. The YME1 is essential both for the degradation of unassembled subunit 2 of cytochrome c oxidase and for efficient assembly of mitochondrial respiratory chain [60]. Autophagy-related protein 11 (*ATG11*) recruits mitochondria for their selective degradation during starvation through its interaction with autophagy-related protein 32 and plays a significant role in life span extension [61]. Mitogen-activated protein kinase HOG1 (*HOG1*) plays a crucial role in the response to various environmental stresses, and mutations in *HOG1* render an organism more sensitive to agents producing reactive oxygen species, such as oxidants and UV light [62]. The *SSK1* product may fulfil regulatory roles in signaling pathways involving a HOG1 MAP kinase during ROS tolerance, osmotic resistance, fungicide sensitivity and fungal virulence [63]. Vacuolar protein sorting-associated protein 1 (*VPS1*) and Atg8 cooperatively participate in vacuolar function, thereby contributing to oxidative stress resistance [64]. Loss of the mitochondrial distribution and morphology protein 38 (*MDM38*) in yeast mitochondria results in a variety of phenotypic effects including reduced content of respiratory chain complexes, altered mitochondrial morphology, and loss of mitochondrial K(+)/H(+) exchange activity [65]. 

#### 3.11.4. Ribosome Formation and Protein Biosynthesis

Cells typically respond quickly to stress and alter their metabolism to compensate for their stress. The nucleolus senses stress and is a central hub for coordinating the stress response, which is fulfilled by rapid production of small and large ribosome subunits, a process that must be highly regulated to achieve proper cellular proliferation and cell growth [66]. Ribosome biogenesis occurs sequentially in the nucleolus, the nucleoplasm and the cytoplasm. It involves the transcription and processing of pre-ribosomal RNAs, their proper folding and assembly with ribosomal proteins, and the transport of the resulting pre-ribosomal particles to the cytoplasm where the final maturation events take place [67]. Several DEGs enriched the ribosome biogenesis. The seven up regulated DEGs (*EMG1*, *UTP14*, *UTP10*, *CKA1*, *FCF1*, *UTP22* and *PWP2*) are all involved in the formation of 90S pre-ribosome in the nucleolus. Casein kinase II subunit alpha (*CKA1*) and U3 small nucleolar RNA-associated protein 22 (*UTP22*) are factors of UTP-C complex; U3 small nucleolar RNA-associated protein 10 (*UTP10*) is a factor of t-UTP complex; periodic tryptophan protein 2 (*PWP2*) is a factor of UTP-B complex. These three complexes participate in the formation of 90S pre-ribosome together with U3 small nucleolar RNA-associated protein 24 (*FCF1*) and U3 small nucleolar RNA-associated protein 14 (*UTP14*) as well as 18S rRNA pseudouridine methyltransferase (*EMG1*). The cleavages of 90S pre-ribosome creates pre-40S and pre-60S ribosomal particles. Then, the both particles are transported out of the nucleolus and into the cytoplasm. Once in the cytoplasm, the pre-60S ribosomal particle further undergoes processing to be functional. To this end, the maturation requires many biogenesis factors. But after cuminal treatment, the 18S rRNA pseudouridine methyltransferase (*MDN1*), large subunit GTPase 1 (*LSG1*) and ribosome assembly protein 1 (*RIA1*) were down regulated. The enrichment of these down regulated DEGs probably made the 60S ribosome subunit not workable, but the precise mechanism remains unclear since pathway of the 60S subunit cytoplasmic maturation is not yet well understood [68].

From transcription to RNA processing and translation, expression regulation of coding genes is controlled at multiple levels. mRNA maturation and translation are post-transcriptional regulatory mechanisms underpinning the genome’s coding capacity in modifying protein function, stability, localization and expression levels [69]. Several genes involved in mRNA maturation and translation were regulated by cuminal exposure. Nuclear cap-binding protein subunit 1(*STO1*) binds co-transcriptionally to the 5’-cap of pre-mRNAs and is involved in the degradation of nuclear mRNAs. *FUN12* encoding the translation initiation factor eIF5B was up regulated after cuminal exposure; eIF5B may be one of the targets among the translation components affected by redox [70]. As to those down regulated DEGs involved in the mRNA processing and translation, they are generally functioning in the translation initiation factors and exon-junction complex. Eukaryotic translation initiation factor 1A (*TIF11*) is required for maximal rate of protein biosynthesis and enhances ribosome dissociation into subunits and stabilizes the binding of the initiator Met-tRNA(I) to 40 S ribosomal subunits [71]. Polyadenylate–binding protein (*PAB1*) appears to be an important mediator of the multiple roles of the poly(A) tail in mRNA biogenesis, stability and translation [72]. ATP-dependent RNA helicase (*FAL1*) is an element of exon-junction complex and responsible for exon splicing. ATP-dependent helicase (*NAM7*) is required for rapid turnover of mRNAs containing a premature translational termination codon [73]. Eukaryotic translation initiation factor 2 subunit gamma (*GCD11*) functions in the early steps of protein synthesis by forming a ternary complex with GTP and initiator tRNA, and the yeast eIF2γ mutation impairs translation start codon selection and thus vitiates translation initiation [74]. Down regulation of these critical genes implies that cuminal exposure impaired the biosynthesis of proteins. 

#### 3.11.5. Cell Cycle and Multiplication 

*T. roseum* reproduces asexually through the formation of conidia with no sexual state [75]. Cuminal exposure altered the transcripts of several genes of the fungus involved in cell cycle pathways, and some of them were up regulated. Among these elevated genes, cell division control protein 45 (*CDC45*) is required for initiation of chromosomal DNA replication and also has a role in minichromosome maintenance [76]. Guanine nucleotide exchange factor LTE1 (*LTE1*) is a GDP-GTP exchange factor for GTP-binding protein involved in the termination of M phase and functions in the mitotic exit network. LTE1 as a signal promotes exiting from mitosis by multiple mechanisms [77]. Structural maintenance of chromosomes protein 4 (*SMC4*) is a central component of the condensin complex required for conversion of interphase chromatin into mitotic-like condense chromosomes [78]. F-box protein MET30 (*MET30*) regulates several aspects of the cell cycle, including G (1)-specific transcription, initiation of DNA replication, and M phase progression [79]. The elevation of such genes suggests that the fungus is on the alert for the possibility of cuminal-caused destruction of chromosomes.

In contrast, more genes involved in cell cycle were down transcripted. multi-functional protein phosphatase PP2A regulatory subunit A (*TPD3*) affects transcription, cell cycle progression, and cellular morphogenesis [80]. RING-box protein HRT1 (*HRT1*) targets Pol II for proteasomal degradation in DNA-damaged cells and thus affects cell division [81]. The anaphase promoting complex/cyclosome activator protein CDH1 (*CDH1*) regulates the ubiquitin ligase activity and substrate specificity of the anaphase promoting complex/cyclosome and is required for exit from mitosis, cytokinesis and formation of prereplicative complexes in G1. Cell cycle serine/threonine-protein kinase (*CDC5*) is involved in mitotic exit, and functions in preserving of genome integrity [82]. Serine/threonine protein kinase (*KCC4*) plays a role in cell wall synthesis and is involved in budding cell bud growth [83]. Structural maintenance of chromosomes protein 2 (*SMC2*) is the central component of the condensin complex required for conversion of interphase chromatin into mitotic-like condense chromosomes [84]. The product of *DBF4* is involved in cell cycle regulation of pre-mitotic chromosome replication and in chromosome segregation. DNA replication licensing factors MCM2 (*MCM2*), MCM4 (*MCM4*), and MCM7 (*MCM7*) act as a component of the MCM complex, which is the putative replicative helicase essential for ’once per cell cycle’ DNA replication initiation and elongation in eukaryotic cells [85]. Cyclin-dependent kinase 1 (*CDC28*) is essential for the initiation, the controlling event, of the cell cycle [86]. Mitotic check point protein (*BUB2*) is a part of a checkpoint monitoring spindle integrity and preventing premature exit from mitosis. Origin recognition complex subunit 1 (*ORC1*) binds origins of replication and has a role in chromosomal replication. Serine/threonine-protein phosphatase PP2A-1 catalytic subunit (*PPH21*) is involved in the control of G1/S transition of mitotic cell cycle [87]. Serine/threonine-protein kinase CHK1 (*CHK1*) is required for checkpoint-mediated cell cycle arrest and for activation of DNA repair in the presence of DNA damage or un-replicated DNA [88]. The down expression of these cell cycle related genes denotes the arresting of fungal cell cycle, precipitating failure of cellular multiplication. 

In proteome, no up regulated protein involved in cell cycle was detected while certain down regulated proteins enriched this cell process. Minichromosome maintenance protein 5 (MCM5) acts as component of a putative replicative helicase essential for ’once per cell cycle’ DNA replication initiation and elongation in eukaryotic cells [85]. Checkpoint serine/threonine-protein kinase (BUB1) is involved in cell cycle checkpoint enforcement and was the most down modulated protein, decreasing nearly 7-fold, in this cellular process. Structural maintenance of chromosomes protein 1 (SMC1) functions in chromosome cohesion during cell cycle and DNA repair. The enrichment of such proteins corroborated the occurrence of cell cycle arresting. 

Additionally, purine salvage is a complex pathway allowing for a correct balance between adenylic and guanylic derivatives; a decrease of the guanylic nucleotide pool connotes cell shifting from proliferation to quiescence. Guanine deaminase, a critical enzyme in purine salvage, was elevated both at transcriptional (*GUD1*) (12.86-fold) and translational (1.78-fold) levels. This enzyme catalyzes the hydrolytic deamination of guanine, producing xanthine and ammonia [89]. Up transcripted 3’,5’-cyclic-nucleotide phosphodiesterase (*PDE1*) catalyzes nucleoside 3’,5’-cyclic phosphate to nucleoside 5’-phosphate [90], suggesting a decrease of the cAMP level. Regulation of intracellular levels of cyclic nucleotides is among the mechanisms involved in cell cycle progression and is of critical importance for cell survival [91].

## 4. Discussion

The essential oil of *C. cyminum* has been extensively explored for its antioxidant and antimicrobial activities [92,93], but most of the investigations focused only on the bioactivities of the essential oil and have seldom considered what is the most responsible component for its specific bioactivity. Although synergy was a popular account of the essential oil bioactivities [94,95], overemphasizing synergy is not useful for the elucidation of the mode of these bioactivities and consequently, for the development of more effective remedies. Since the preciousness of natural-derived essential oils makes it financially unfeasible to practically use them to achieve the desired effects, the investigation of the function mechanisms of their bioactive components is of practical significance, so that some low cost alternatives can be developed. Our earlier research has shown that cuminal is the most predominant component of the *C. cyminum* essential oil and acidolysis pretreatment can elevate the contents of cuminal and the antifungal activity of the essential oil [26]. Indeed, the antifungal activity of the individual cuminal was twice as strong as that of the essential oil itself, suggesting that intensive examination of the function mode of cuminal against fungal growth will be conducive to the interpretation of the antifungal mechanism of cumin essential oil. 

As the predominant component of the cumin essential oil, lipophilic cuminal molecules should have strong affinity to cytoplasm membrane; thus, cuminal exposure elicited the fungal responses from the membrane first. When intruding onto the lipid bilayer of cytoplasm membrane, cuminal damaged the polarity of the cytoplasm membrane. Membrane depolarization is associated with the membrane fluidity decrease and compromised functionality [96]. In the present investigation, cuminal exposure rendered the fungal transcription of genes functioning in membrane lipids metabolism up regulated. These up regulated genes mainly collaborate in the biosynthesis of phosphatidylcholines, inositol phosphates and phosphatidylethanolamines. In contrast, the down regulation of cardiolipin synthase suggests the limited cardiolipin biosynthesis. Both transcriptome and proteome investigations showed the elevation of the biosynthesis of phosphatidylethanolamines, but the reason for such an elevation was not reported. Additionally, a number of eukaryotic host defense peptides such as plant cyclotides use phosphatidylethanolamines as a receptor to promote their antimicrobial activities [97]. These changes of membrane lipid biosynthesis may alter the ratios of various lipid species of the membrane; the resulting lipid profile changes might impact the interactions among these molecules and so induce membrane disorder [98]. Fungal plasmalemma is a universal target explored extensively for the development of antifungal agents. This strategy has been proved fruitful by the pronounced success of antifungal drugs such as azoles and polyenes [99]. Consistent with the down regulation of most of membrane phospholipid synthesis was the overall trend of ergosterol biosynthesis of the membrane. Ergosterol is the primary sterol in fungal membranes and presumably contributes to membrane fluidity and function [100,101]. The fungus responded to cuminal exposure with the down regulation of the ergosterol biosynthesis genes (*ERG25*, *ERG9*, *ERG6*, and *ERG5*), and this regulation was consistent with the decrease in ergosterol content of the cellular membrane (data not published). By increasing the permeability of membrane, modulating the activity of membrane-bound enzymes in the plasmalemma and mitochondria, stimulating uncoordinated chitin synthesis or interfering with fatty acid synthesis, a deficiency in ergosterol affects fungal viability and growth [102]. Furthermore, and interestingly, cuminal exposure strengthened the biosynthesis of polyunsaturated fatty acids. This positive adaptation of the fungus to the adverse conditions is common in other stress threatened microbes [103,104]; the up regulated desaturation is a cellular response to environmental stresses, protecting cells from toxic oxygen species and other detrimental factors [105].

ABC transporter proteins are crucial for pleiotropic drug resistance, stress response and cellular detoxification of fungi [106]. Cuminal exposure elevated the gene expression of these kinds of proteins in both transcriptome and proteome. This elevation might be instinct protection against adversities, but this endeavor seems futile in conquering the overwhelming intrusion of cuminal. Accordingly, cuminal intrusion into the cytoplasmic membrane begot further cellular defensive responses and resulted in dysfunction of the membrane.

Another important consequence of cuminal exposure was the occurrence of oxidative stress (OS) upon the fungus because environmental toxic agents can induce the level escalation of ROS in aerobic organisms surrounded by such agents [46]. OS results when production of ROS exceeds the removing of these toxic species by cellular antioxidant systems, and some important systems of such involve special cellular organelles and antioxidant enzymes including superoxide dismutase, catalase, and peroxidases, which detoxify the cells of harmful ROS, thereby reducing damage to the cells [107]. Peroxisomes are highly dynamic and metabolically active organelle playing a critical role in regulating cell perception and fast responses to environmental cues of stresses [108]. Accumulating evidence indicates that peroxisomes are of primary importance during the maintenance of the cellular oxidative homeostasis [109], and, in the present investigation, up regulation of certain key genes functioning in oxidative defense might enhance the fungal tolerance to cuminal. The failure of sustaining supply of reducing power by the collapsed carbohydrate metabolism, meanwhile, aborted such adaptive attempt in facing the persistent oxidative attacks of cuminal exposure. 

Autophagy, a catabolic process for recycling cellular components and damaged organelles, possibly occurs when cells are under diverse stress conditions. Acting as the universal and converging stress directly from some oxidative chemicals or derivatively from some adversities such as starvation or poisoning, OS induces sustaining autophagy [110]. When it properly occurs, mitophagy, a selective autophagy of mitochondria during the maintenance of cell homeostasis, can mitigate the oxidative stress and protect cells from noxious ROS [111,112]. The process of mitophagy plays multimodal roles in the survival of cells. Benign mitophagy renders the cell adaptive to certain levels of stress, but under lasting or overwhelming stress, abnormal mitophagy takes place and compromises the cells viability [113,114]. Mitochondria play a pivotal role in the energy acquisition of eukaryotes and too great a loss of the crucial organelle destroys the energy metabolism of an organism. The escalation of mitophagy of *T. roseum* upon cuminal exposure might be one aspect of mechanism underpinning the drug inhibition.

As mentioned, cuminal-instigated disturbance in the cytoplasmic membrane component profiles would trigger the dysfunction of the membrane and destroy the nutrient uptake, thus causing the nutrition deprivation to the fungus. This deprivation diminished the central carbon metabolism flux of the fungus. Integrated transcritome and proteome analysis showed the glycolysis was generally down regulated particularly due to the low expression of the *HXK1* (encoding hexokinase-1) and the CDC19 (pyruvate kinase 1), since both of the enzymes are speed-limit ones to the glycolysis. The down regulation indicates the fungal cell carbohydrate starvation resulting from the dysfunction of cytoplasmic membrane. For surviving, the cells have to draw on the other interconnected metabolites to compensate for this carbohydrate deprivation. Gluconeogenesis is a typical way to replenish the carbohydrates and is sensitive to glucose deficiency [115]. Up regulation of gluconeogenesis could transiently refuel the cells while consuming precursor metabolites derived from other metabolism and finally resulting in the comprehensive starvation of the cells. Long term starvation renders the wild type *Neurospora crassa* more sensitive to heat shock and oxidative stress and brings about lethality to the fungus [116]. In the present investigation, gluconeogenesis experienced more sophisticated regulation. Although certain genes were up regulated at transcription level, the down regulation of *PCK1* at both mRNA and protein levels indicates that more intermediate pyruvate might be diverted to the citrate cycle, which further proved the glucose starvation of the fungus. The overall down regulation of the oxidative phase of pentose phosphate pathway indicated that NADPH production declined. Lack of the NADPH crippled the anabolic reactions for the biosynthesis of nucleic acids and lipids and weakened the oxidative stress defense, thereby reducing the growth or viability of the fungus [117].

Opposite to the regulation of glycolysis was the elevation of the citrate cycle under the treatment of cuminal. The overall positive modulation of citrate cycle of the *T. roseum* seems at odds with the down regulation of the glycolysis, whereas, in fact, the active β-oxidation and strengthened degradation of val, leu, Ile and other amino acids could fuel the citrate cycle with acetyl CoA, which accounts for the elevation of citrate cycle. Furthermore, carbohydrate deprivation promotes the lipid droplets-mitochondria interaction and lipid droplets efficiently supply fatty acids for mitochondrial β-oxidation [118]. The products of β-oxidation, especially acetyl-CoA, feed the citrate cycle of central carbon metabolic pathways [119]. Moreover, acute carbohydrate deprivation induces autophagy and elevates amino acid catabolism, which helps maintain transient homeostasis of cells in nutrient scarce conditions to facilitate their viability [120]. 

A universal consequence of the nutrient deprivation was the failure of bioenergetic system of cells. Cuminal exposure of *T. roseum* down regulated certain components of respiratory chain complex II, III, IV and V, and thus crippled the oxidative phosphorylation of the fungus. These modulations on respiratory chains manifested mitochondrial stress resulting in reduced electron transport efficiency and cellular energy deficiency [121]. Also, decreases in respiratory chain complex activities are thought to be associated with oxidant/antioxidant imbalance and induce cellular degeneration [122]. Considering that mitochondria are the primary source of ROS, less efficient respiratory chains means that more ROS are generated, which exacerbates the oxidative stress of cells [123]. Cumulatively, these deleterious effects could eventually cause cellular oxidative damage and beget the loss of cell viability. 

Under the deprivation of carbohydrate and resultant energy deficiency, the biosynthesis of other cellular components might be inevitably vulnerable. Cuminal exposure effectuated diminished central carbon metabolism and incapacitated the supplies of enough precursors, energy, and reducing powers for the anabolism of essential components for the cell survival and proliferation. In present investigation, an active modulation of ribosomal formation was observed. More up regulated genes enriched the pre-ribosome formation, and this might be an attempting protection against the abiotic stress, but very little is known about this sophisticated process. Nonetheless, several genes, such as *MDN1*, *LSG1* and *RIA1*, being pivotal to ribosome maturation, were down regulated, which denotes the failure of providing sufficient efficacious translational apparatus. Furthermore, a notable amount of genes functioning in the translation initiation and mRNA maturation were down regulated, further impairing the protein biosynthesis.

Eukaryotic cells have developed sophisticated systems to constantly monitor changes in the extracellular environment and to orchestrate proper cellular responses so as to accomplish stress adaptation. To maximize survival, cells delay cell-cycle progression in response to environmental insults [124]. Activation of stress responses can induce diverse physiological changes, including modulation of cell cycle progression [125] and excessive stress on replication occasions mitotic cell death [126,127]. In line with the above statements, after cuminal treatment, an overall down regulation of the genes functioning in DAN replication, cell cycle and cell proliferation was observed. This was not surprising, because the energy deprivation of the fungus begot a deficiency of the nucleotides, the raw materials of nucleic acid biosynthesis, which precipitates stress on cell replication [128] of the fungus. Intense stress instigates the transition of cells from growth to quiescence, which is often accompanied with cell cycle modulation, housekeeping function down-regulation, and fierce metabolism changes, all as strained protections against stress [129]. The deactivation of cell replication involves stepwise physiological changes with an intermediate state of being incapable of initiating replicative processes but still capable of metabolism, and this loss of replication competency eventually leads to cell death [130]. All the fungal responses, aforementioned, to cuminal exposure could be delineated on a logical diagram of proposed mode of action to conclude the antifungal mechanism of cuminal against *T. roseum* (Figure 12).

## 5. Conclusions

Cuminal inhibited the growth of *T. roseum* in vitro and in vivo. Electron microscopic observations disclosed the inhibition is related to the degenerating of cellular ultrastructure, especially the plasmalemma. Cuminal exposure regulated *T. roseum* gene expression at both transcriptome and proteome levels. Totally, omics analysis determined 2825 differentially expressed transcripts (1516 up and 1309 down) and 225 differentially expressed proteins (90 up and 135 down). Overall, notable parts of these DEGs functionally enriched subcellular loci of membrane system and cytosol, along with ribosomes, mitochondria, and peroxisomes. In line with the locality analysis, carbohydrate and lipids metabolism, redox homeostasis, and asexual reproductive were among the most enriched GO terms in biological annotation of these DEGs. The up regulated genes more enriched the lipids degradation and antioxidant responses, and the down regulated ones, in contrast, enriched the carbohydrate catabolism, energy acquisition, cellular reproduction, and ribosomal functions. Moreover, the KEGG pathway enrichments of the DEGs embraced elevated lipids and amino acids degradation, ATP-binding cassette transporters, membrane reconstitution, mRNA surveillance pathway, and peroxisome, along with the diminished secondary metabolite biosynthesis, cell cycle, and glycolysis/gluconeogenesis. Furthermore, integrated omics analysis demonstrates that cuminal first impaired the functional integrity of cytoplasmic membrane and triggered the reconstitution and dysfunction of membrane resulting in handicapped nutrients procurement of the cells. Consequently, cell starvation occurred, and cellular carbohydrate metabolism was limited, and the cells might have to depend more on the degradation of their own components in response to the stress. Additionally, these predicaments occasioned oxidative stress, which, in collaboration with the starvation, damaged certain critical organelles such as mitochondria. Such degeneration together with energy deficiency suppressed the biosynthesis of essential proteins and paralyzed the cell multiplication.

## Figures and Tables

**Figure 1 microorganisms-08-00256-f001:**
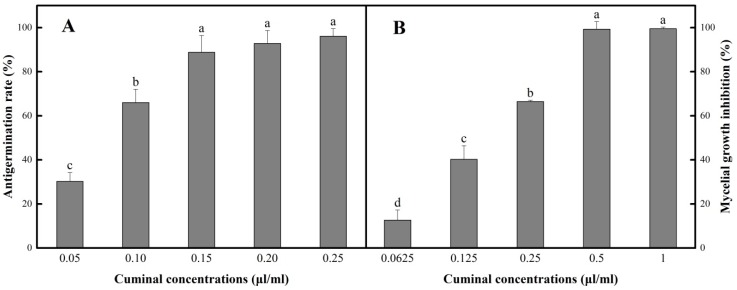
In vitro antifungal activities of cuminal against *T. roseum*. (A) Antigermination effect against *T. roseum* conidia. (B) Vapor contact inhibition of mycelial growth. Bars with different letters indicate mean values significantly different at *p* < 0.05 according to Tukey test. Data are expressed as mean of three replicates ± SD.

**Figure 2 microorganisms-08-00256-f002:**
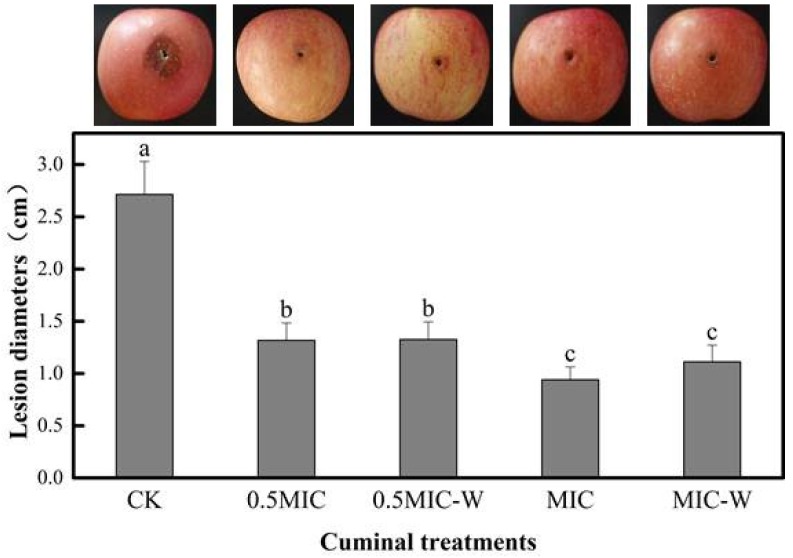
In vivo antifungal effect of cuminal against *T. roseum* on apple fruits. Bars with different letters indicate mean values significantly different at *P* < 0.05 according to Tukey test. Data are expressed as mean of three replicates ± SD.

**Figure 3 microorganisms-08-00256-f003:**
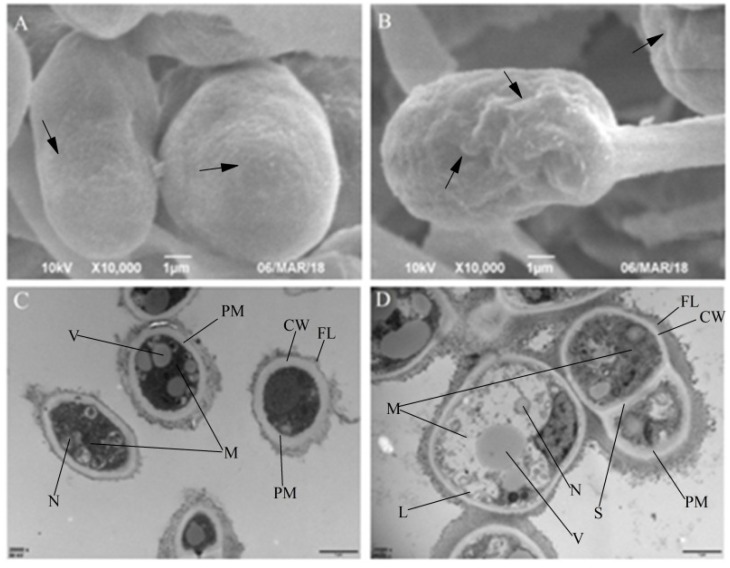
Morphological (**A**: control, and **B**: treatment under scanning electron microscopy) and ultrastructural (**C**: control, and **D**: treatment under transmission electron microscopy) changes of *T. roseum* conidia upon cuminal exposure. (CW: cell wall, FL: fibril layer, PM: plasmalemma, M: mitochondrion, S: septum, V: vacuole, N: nucleus, L: lomasome).

**Figure 4 microorganisms-08-00256-f004:**
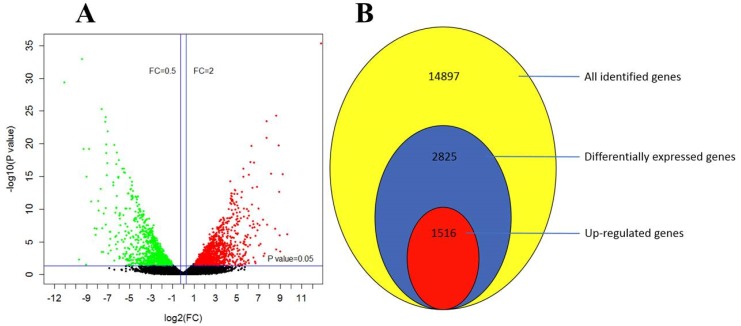
Cuminal exposure regulated gene expression of *T. roseum* at mRNA level. (**A**) The volcano plot of detected genes indicating significantly up (red) and down (green) regulated genes. (**B**) Comparison of the numbers of all identified genes and differentially expressed genes (up vs. down regulated).

**Figure 5 microorganisms-08-00256-f005:**
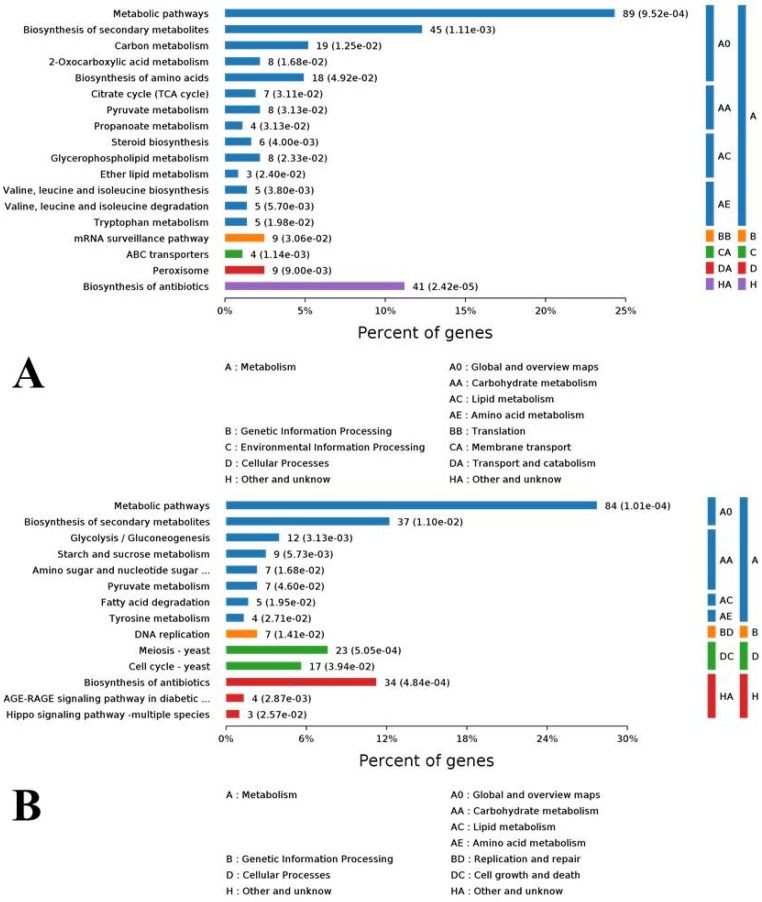
Kyoto Encyclopedia of Genes and Genomes (KEGG) enrichment of differentially expressed genes (DEGs) and hierarchical belonging of each enriched pathway to upper level pathways. (**A**) Pathway classification of up regulated DEGs. (**B**) Pathway classification of down regulated DEGs.

**Figure 6 microorganisms-08-00256-f006:**
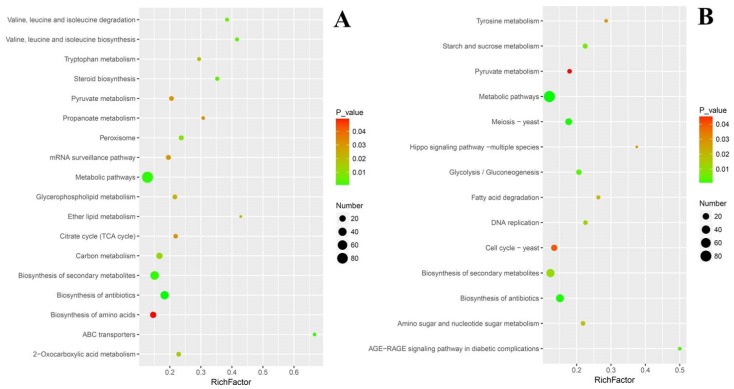
Rich-factor scatter plot of select pathway enriched with DEGs. (**A**) Up regulated DEGs. (**B**) Down regulated DEGs. Rich factor is the number ratio of differentially expressed genes to total annotated genes in a pathway. Dot areas show the number of DEGs, and the selection was based on the KEGG enrichment situation and pertinence to the metabolic blocks underpinning the fungal response to cuminal exposure.

**Figure 7 microorganisms-08-00256-f007:**
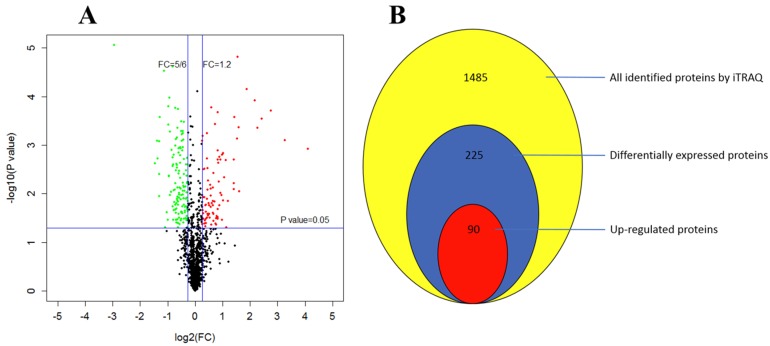
Cuminal exposure regulated gene expression of *T. roseum* at protein level. (**A**) The volcano plot of detected proteins indicating significantly up (red) and down (green) regulated proteins. (**B**) Comparison of the numbers of all identified proteins and differentially expressed proteins (up vs. down regulated).

**Figure 8 microorganisms-08-00256-f008:**
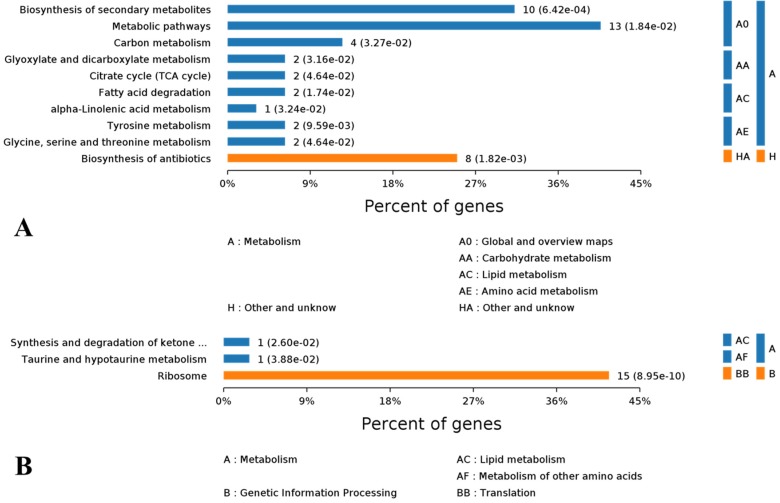
KEGG enrichment of DEPs and hierarchical belonging of each enriched pathway to upper level pathways. (**A**) Pathway classification of up regulated DEPs. (**B**) Pathway classification of down regulated DEPs.

**Figure 9 microorganisms-08-00256-f009:**
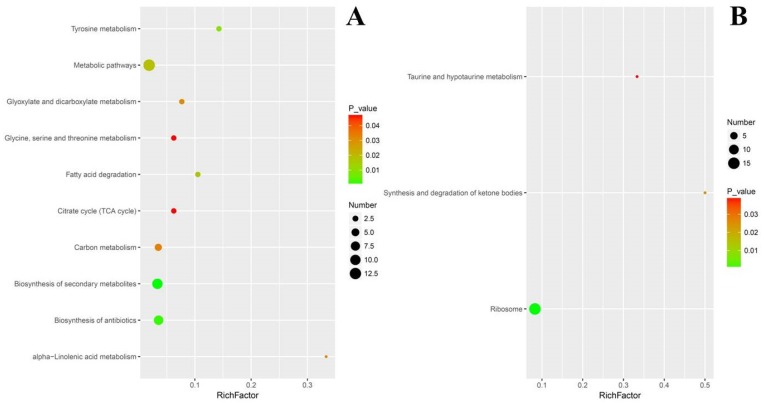
Rich-factor scatter plot of select pathway enriched with DEPs. (**A**) Up regulated DEPs. (**B**) Down regulated DEPs. Rich factor indicating the number ratio of differentially expressed proteins to total annotated proteins in a pathway. Dot areas show the number of DEPs, and the selection was based on KEGG enrichment situation and pertinence to the metabolic blocks in the fungal response to cuminal exposure.

**Figure 10 microorganisms-08-00256-f010:**
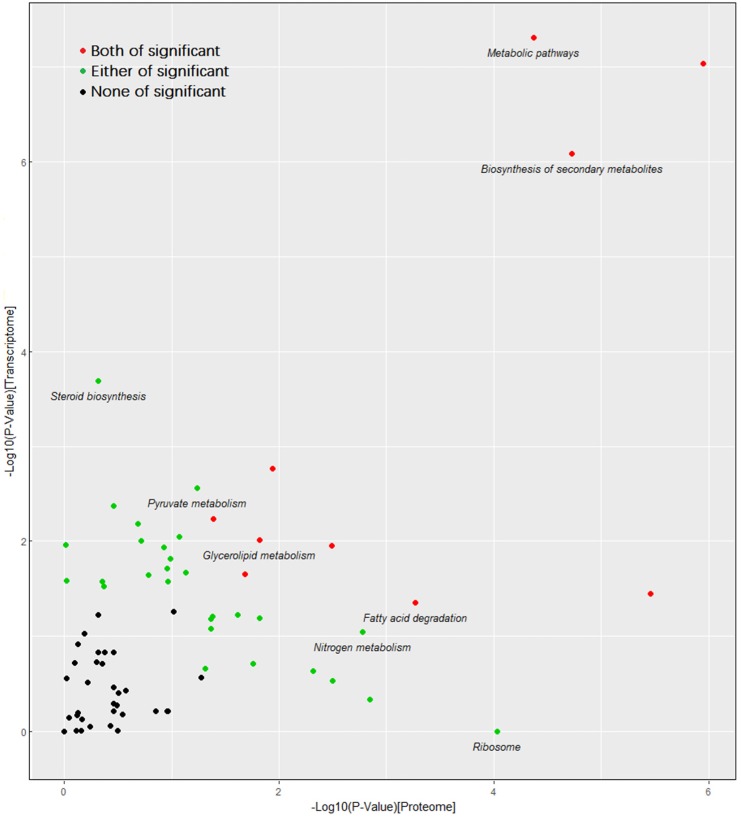
Scatter diagram of KEGG enrichment correlation between the datasets of proteome and transcriptome.

**Figure 11 microorganisms-08-00256-f011:**
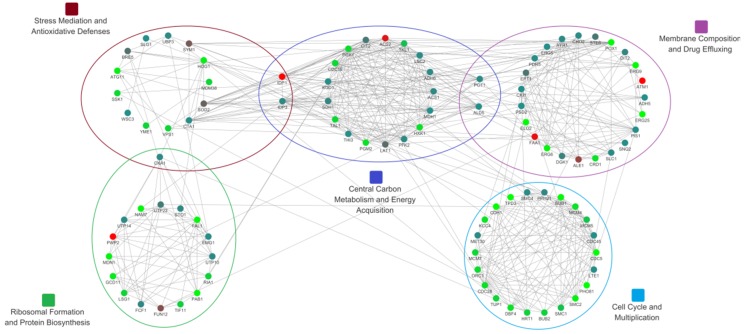
Protein–protein interaction network of *T. roseum* under cuminal exposure compared with the control. The proteins encoded by DEGs functioning in the fungal metabolism discussed in the context were selected. Interactions are shown by the lines connecting each node based on available evidence in the database. Interactions are integrated into different metabolic blocks indicated by the colored circles.

**Figure 12 microorganisms-08-00256-f012:**
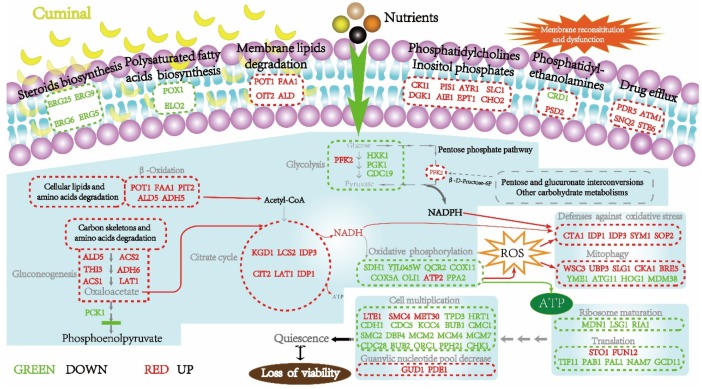
Proposed mode of action of cuminal against *T. roseum*. Red color in the diagram shows up regulation of genes or overall elevation of the indicated metabolic blocks while green color indicates down regulation of genes or overall suppression of a metabolic bock. The metabolism blocks that were deeply analyzed in the text are circled with red or green dash lines.

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
