# Peer review of "Cuminal Inhibits Trichothecium roseum Growth by Triggering Cell Starvation: Transcriptome and Proteome Analysis"

_microorganisms, 2020, doi:10.3390/microorganisms8020256_

Round 1

Reviewer 1 Report

The major weakeness of the manuscript is the lack of a proper investigation of the essenzial oil effects on the growth, development and virulence of the fungal pathogen. I suggest to implement the study adding further data on the in vitro mycelial inhibition due  to cumin essential oil and on the disease control.

Moreover, authors should explain why random real Time PCR esperiments has not performed to check bioinformatic findings.

Please, add the required data to improbe the manuscript and to link molecular findings with the real phenotype

Reviewer 2 Report

The manuscript entitled “Transcriptome and Proteome Analysis Reveals Cuminal Inhibits Trichothecium roseum Growth by Triggering Cell Starvation” aims to evaluate the action of fungus Trichothecium roseum to essential oil Cuminum cyminum by transcriptome and proteome analysis. Fruits and vegetables are easily deteriorated by fungal diseases, resulting in quality loss and severe losses of produce. Due to concern about the fungicide efficacy and resistance, new substances need to be evaluated, being the essential oil a promise, alternative, natural with antimicrobial activity. The understanding of fungal biology under exposure of the principle bioactive component of essential oils could achieve the attention about the antifungal essential oils. The main goal of the study it is relevant and understanding the mechanisms of antifungal can be a efficient functional targets and improve the culminal application for achieve new technologies, economical and sustainable solutions to control the fungal disease in fruits. The choice of methodologies was adequate for purpose of study.
Some observations addressing the sections:
In page 5, line 220 – “…(Figure S2). -please change the number of figure.
In conclusions section, from line 860 to line 868, the sentences are quite similar of observed in the abstract section. The authors indicated the main results and conclusions of their study, however a re-formulation of these parts need to be done for quality improvement of the manuscript.

Round 2

Reviewer 1 Report

Thanks a lot, all requirements have been received. The manuscript can be considerate for publication on this journal.